# Production of Biodiesel and High-Protein Feed from Fish Processing Wastes Using In Situ Transesterification

**DOI:** 10.3390/molecules25071650

**Published:** 2020-04-03

**Authors:** Tongdong Zhang, Beiyan Du, Yuexu Lin, Min Zhang, Yueliang Liu

**Affiliations:** 1School of Life Sciences, Fujian Agriculture and Forestry University, Fuzhou 350002, China; dby91014@163.com (B.D.);; 2Key Laboratory of Unconventional Oil & Gas Development, China University of Petroleum (East China), Ministry of Education, Qingdao 266580, China; 3School of Petroleum Engineering, China University of Petroleum (East China), Qingdao, Shandong 266580, China

**Keywords:** fish, feed, biomass, biodiesel, co-solvent, in situ, fatty acid, transesterification

## Abstract

Preparation of biodiesel using in situ transesterification has been extensively conducted for agricultural, microbial and algal biomass, while few works have been performed using aquatic animal tissue. In this work, fish processing wastes were collected to perform in situ transesterification using grass carp (*Ctenopharyngodon idellus*) biomass as a representative with which to optimize the reaction conditions. Under the optimum condition, the highest biodiesel purity reached up to 100% for sea bass wastes, which is higher than the 96.5% specified in the EN 14214-2008. The in situ method proposed here has the potential to save significant costs in biodiesel production compared to conventional methods, which usually require high-cost pretreatment of the raw materials. Additionally, the waste residue byproduct produced has a high protein content, and therefore the potential to be used for high-protein feed. This study is expected to inspire new strategies to prepare biodiesel and high-protein feed simultaneously from aquatic animal biomass using the novel in situ transesterification.

## 1. Introduction

Fish fat is one of the main wastes produced internationally from international aquatic products. In 2015, fish production in the world reached about 166.8 million tons [1]. However, more than 50% of the production was discarded rather than recycled, representing a challenge for environmental management; these highly oily wastes can be used as the raw materials for biodiesel production [1], replacing edible oil raw materials (such as soybean oil) to ensure food security [2,3].

Transesterification is one of the most widely used methods for biodiesel production, and is generally clarified into two categories, i.e., conventional and in situ transesterification. The raw materials used in conventional methods must undergo pretreatments such as oil extraction and refining processes (e.g., degumming, deacidification, decolorization, deodorization, dewaxing). However, in situ transesterification can utilize untreated (or pre-dehydrated) oil-rich biomass directly, and the chemical reaction usually occurs in the mixture of oil and solids [4]. Extensive studies have mentioned that the cost of oil extraction and refining in traditional production processes are very high, accounting for 70–80% of the total cost of biodiesel production [4,5]. This is a possible reason that the price of biodiesel is higher than that of petro-diesel. Therefore, the in situ method may be an alternative that could reduce the total cost of biodiesel production.

The in situ conversion method was first proposed by Dugan [6], and has received a widespread attention since 2007 [4]. To date, extensive works regarding the preparation of biodiesel via the in situ transesterification method using different raw materials have been reported [4]. These works have mainly focused on various plant materials and fungi, etc. Although plenty of studies have reported conventional transesterification using fish raw materials, the in situ method has never been applied to produce biodiesel from aquatic animal biomass. It should be noted that the biomass of fish processing wastes is quite different from that of plants because the chemical composition of fish tissues is more complicated, and the acid value of waste fat is much higher. Techniques that are commonly used for the production of non-animal biodiesel (such as alkali-catalyzed transesterification) generally struggle to efficiently convert fish waste fat with a high acid value.

The cost of the raw materials comprises 60–70% of the total cost of biodiesel production processes [7]. An effective way to reduce the cost of biodiesel production is to use raw materials which are high-yield, high-oil, but low-priced, such as waste fish tissue. Study has shown that blends of fish oil and diesel fuel can produce decreased emissions of up to 60% of particulate matter (PM), 33% of CO, and 78% of SO_2_ emissions compared to traditional diesel fuel [8]. Waste fish oil has great potential for biodiesel production, and can be used as an alternative to traditional raw materials; however, research into these production strategies is still at an early stage [9]. Previous works [10,11] have investigated biodiesel production from fish oil at an industrial scale; it was found that the extraction and purification of the waste of fish oil are the key challenges limiting the large-scale production of biodiesel. It is worth mentioning that the alkaline catalyst technology, which is widely used in factories, involves several complex extraction and purification steps, while the final biodiesel conversion rate is less than 30%. Therefore, this novel in situ transesterification technology, which has not been applied to aquatic animal biomass until now, seems to be a cost-effective solution. Another benefit of the in situ technology is the possible production of animal feed as a byproduct, which could further reduce the cost of biodiesel production and even lower the price of commercially available feed. At present, research in this area has only applied a very few plant materials, such as soybean [12], rapeseed [13], and cottonseed [14].

Grass carp is one of the “four major domestic fishes” in China, and is also the highest-produced farmed fish in the world. Although it has been introduced to more than 100 countries [15], research on the preparation of biodiesel from such fish oil is still rare. In this study, grass carp processing waste was used as an example of fish biomass to investigate the optimal reaction conditions, and sulfuric acid was used as a catalyst. The optimal reaction conditions were determined for the application of the in situ transesterification method to transform fish tissue with high acid value into biodiesel. To some extent, our research overcame the challenge associated with the low quality (e.g., high free fatty acid content, solid impurities, waxes, and pigments) of pre-dehydrated aquatic animal biomass. Finally, we presented a reasonable proposal that could be applied to both freshwater and marine fish tissues to directly prepare high-purity biodiesel. However, thanks to the use of organic solvents during the preparation, the obtained crude biodiesel may still contain some impurities (e.g., free fatty acids, glycerin, and glyceride) and need further purification. To our knowledge, this is the first time that the reaction conditions for application of the in situ transesterification method to produce biodiesel and animal feed simultaneously for fish biomass, or even for animal biomass.

## 2. Results and Discussion

We first screened the optimal co-solvents before the experiment. The effects of chloroform, n-hexane, and acetone as co-solvents in the in situ transesterification of grass carp were investigated under the same reaction conditions. It was found that the use of n-hexane as a co-solvent increased the conversion rate of the reaction more than other co-solvents, which was consistent with H_2_SO_4_ as catalytic in the conventional transesterification using the carp oil [16].

### 2.1. Fatty Acid Profile of Fish Fats

The Fatty acid methyl ester (FAME) composition of biodiesel depends on the raw material used in its synthesis. The fatty acids of various types of oils and fats are different and play a decisive role in the properties of biodiesel products. We compared the fatty acid compositions of the fats from three different fish tissues used in this study, and lard was used for comparison, as shown in Table 1. Among these fats, C16: 0, C18: 1, and C18: 2 were the majority fatty acid components.

The content of saturated fatty acids (SFA) and monounsaturated fatty acids (MUFA) is generally used to evaluate the oxidative stability of fats and oils. We found that the total content of SFA and MUFA in the fat of freshwater fish like grass carp and catfish was as high as 79.19% and 72.97%, similar to lard (73.87%), which is generally used for biodiesel feedstock with relatively high oxidation stability. Moreover, a study on carp oil found that the SFA+MUFA content was 89.63% [16]. These results show that such freshwater fish oils are highly stable biodiesel feedstocks. By contrast, the polyunsaturated fatty acid (PUFA) content of marine fish tissues like sea bass fat is significantly higher, reaching 43.14%, indicating that the oxidative stability of this type of marine fish oil is relatively low, while it suggests a good tolerance to low temperatures. This type of biodiesel is especially suitable for use in areas where temperatures are low throughout the year.

### 2.2. Effects of Reaction Factors on Conversion Efficiency of Biodiesel

#### 2.2.1. Ratio of co-Solvent to Methanol Volume

The introduction of the proper amount of co-solvent can improve the reaction efficiency and reduce the amount of methanol. We initially referred to the co-solvent dosage from the conventional method for the preparation of fish oil biodiesel [16] and the in situ method for preparation of biodiesel from plant biomass [17], employing a volume of n-hexane higher than or equal to the dose of methanol, for which the conversion efficiency was extremely low. The specific experimental results are shown in Figure 1. This may have been due to the low quality of the fish tissue, such as the presence of solid impurities. When the proportion of methanol increased, the conversion efficiency gradually improved at the same time. The conversion efficiency reached the highest point when n-hexane: methanol was 1: 8.87. As the volume ratio of methanol further increased, the conversion efficiency was reduced because the solubility of fats in methanol is extremely low, so that they could not be completely extracted and reacted. Nguyen et al. [18] determined the optimal n-hexane: methanol volume ratio, which was determined to be 1: 2 for the in situ transesterification of biodiesel from the black soldier fly larvae. However, our experimental results disagreed with this. The difference may have been caused by the disparity in the chemical composition of distinct raw materials. Additionally, a tissue homogenate was used in this study, in contrast to the biomass particles used in the experiment of Nguyen et al. [18]. The lipids contained in a homogenate are more likely to directly and efficiently react with methanol. Therefore, the ratio of co-solvent used was depressed. In order to achieve the maximum conversion efficiency and simplify the calculation, we selected a 1: 9 ratio of n-hexane: methanol (*v/v*) in the subsequent experiments.

#### 2.2.2. Solvent Dosage

Figure 2 presents the effect of solvent dosage (hexane + methanol) on the conversion efficiency. As the solvent dosage increased from 7.4 mL to 31.0 mL, the conversion efficiency of the in situ transesterification increased from 9.12% to 36.08%. Upon further increasing the solvent dose to 35.0 mL, the conversion efficiency dropped to 30.44%. This was because it was difficult for the low solvent dose to thoroughly extract the oil from the biomass; thus, the reaction rate was slow, and the reaction time was prolonged. However, excessive solvent dosage reduces the frequency of collisions between fat and methanol, which increases the heat and mass transfer resistance and thus decreases the conversion efficiency [19,20]. As a result, we chose the solvent dosage of 31.0 mL.

#### 2.2.3. Catalyst Loading

Sulfuric acid, a widely used catalyst in transesterification, can simultaneously catalyze the conversion of free fatty acids and glycerides to FAME without the occurrence of saponification reactions. Theoretically, it was particularly suitable for the in situ transesterification of high acid value biomass in this experiment. Therefore, we chose H_2_SO_4_ as our catalyst. Figure 3 shows the effect of different doses of sulfuric acid on the conversion efficiency. As the sulfuric acid dose was increased from 0.3 mL to 0.6 mL, the biodiesel conversion rate was improved 2.5-fold. Upon continuing to increase the amount of sulfuric acid, the conversion efficiency decreased. It is believed that excessive addition of sulfuric acid will exacerbate the oxidation of unsaturated fatty acids [21]. Liu et al. [22] used coffee grounds to prepare biodiesel using the in situ method; it was found that the biodiesel yield increased continuously with the increasing sulfuric acid concentration in the solution, which was a different result from our experiment. This may have been because their research set a small range of sulfuric acid concentration gradients that could not reflect the effect of excess sulfuric acid dosage on the biodiesel yield. In this experiment, the optimal sulfuric acid dosage was set as 0.6 mL.

#### 2.2.4. Reaction Temperature

We chose the temperature gradients of 65 °C, 75 °C, and 85 °C to investigate the effect of temperature on the conversion efficiency. The experimental data are shown in Figure 4. We found that when the reaction temperature was increased from 65 °C to 75 °C, the biodiesel conversion efficiency was increased by about 20%; in addition, when temperature further increased to 85 °C, the conversion efficiency decreased by 10%. This indicated that the reaction temperature is a key factor in affecting the in situ transesterification of fish tissues. Excessive temperature not only increased the energy input of the biodiesel production process, but also increased the evaporation of the solvent and accelerates the oxidation of the oil, consequently reducing the conversion efficiency of biodiesel. The optimum temperature was set as 75 °C in this study, which was significantly lower than were 90 °C and 120 °C temperatures used by previous studies focusing on microalgae [19] and black soldier fly larvae [18], indicating that the technique established here could effectively decrease the energy cost invested in the biodiesel preparation process and reduce the performance requirements of the reaction machine in the biodiesel production industries. We thus chose the reaction temperature of 75 °C for the optimization of the next reaction condition.

#### 2.2.5. Reaction Time

For reversible transesterification reactions, time is an important factor affecting the conversion efficiency. A transesterification reaction cannot reach equilibrium within only a short reaction time, while a long reaction time not only increases the energy input but also represses biodiesel yield due to the high concentration of the product held in the reaction system. For the optimization of the reaction time, six reaction times, i.e., 1.0, 1.5, 2.0, 2.5, 3.0, and 4.0 h, were selected. The experimental results are shown in Figure 5. It was obvious that the conversion efficiency of the in situ transesterification achieved its highest point at 2.5 h with a conversion efficiency of 75.19%. In addition, increasing the reaction time gradually decreased the biodiesel conversion efficiency. The optimal reaction time proposed here was drastically lower than 12.0 h reaction time suggested by Liu et al. [22]. A small amount of co-solvent was introduced in this study, which may have accelerated the dissolution of the oil contained in the biomass and significantly shortened the reaction time [23].

### 2.3. Property of the Product, Byproduct, and Raw Materials

Acid value is one of the most important indicators for evaluating the properties of biodiesel. Dias et al. [24] reduced the acid value of lard from 14.57 to 3.00 mg KOH g^−1^ using a 3 wt.% H_2_SO_4_-catalyzed transesterification reaction. The acid value of the fish oil contained in the grass carp tissue used in our research was as high as 19 mg KOH g^−1^ initially. After the in situ transesterification under the optimal conditions, the crude biodiesel had an acid value of only 1.0 mg KOH g^−1^, indicating a certain improvement over some conventional transesterification methods. The biodiesel obtained may have greater competitiveness after the subsequent industrial distillation and purification measurements.

Qian et al. [13,14] mentioned that in the process of in situ transesterification of rapeseed and cottonseed, the crude protein content was increased by 15.5% and 18.6%, respectively. In this study, the crude protein content of the original grass carp tissue was only 4.38%, due to its high fat content. After the reaction of the in situ transesterification under the optimal conditions, it was found that the crude protein content was increased by 30.35% in the reaction wastes, reaching 34.73%. The result implies potential of the byproduct obtained as a high-protein animal feed, which would not need to undergo a process of removing the toxic substances (such as glucosinolates and gossypol) contained in the plant materials used in the previous studies.

After investigating the biodiesel synthesized from grass carp, we paid attention to the yield of grass carp tissue materials. The wet weight of the raw materials used in this study accounted for about 11% of the total weight of a grass carp. According to Food and Agriculture Organization of the United Nations (FAO) in 2017 [25], grass carp aquaculture production has reached 5.52 million tons. Consequently, grass carp waste biomass alone could reach more than 607,200 tons. Furthermore, it is worth emphasizing that the aquatic animal tissues used in this study are one of the most oil-containing raw materials (e.g., the oil content of grass carp dry tissue can reach 87.4%). Additionally, fish waste tissue has the advantage of extremely low price [2]. As a result, it is believed that the proposed in situ transesterification method of biodiesel synthesis using waste fish tissue is economically and commercially viable.

### 2.4. In Situ Transesterification of Other Fish Tissues

In order to verify the applicability of the in situ transesterification technology optimized for grass carp tissue, we carried out in situ transesterification under the optimal reaction conditions using the tissues and viscera from freshwater catfish and sea bass. The biodiesel conversion rate and purity are shown as indexes in Table 2. It was found that the in situ transesterification method optimized using grass carp could still achieve a high biodiesel conversion efficiency and purity when applied to other kinds of aquatic animal materials. Crude biodiesel prepared from the biomass of sea bass can exceed the purity specified in EN 14214-2008. 

Previous works have also investigated the properties of synthesized biodiesel. For example, Dias et al. [24] employed a relatively pure material of a mixture of lard and soybean oil, and biodiesel, which was prepared via acid–base two-step reaction, which yielded only 64.4% FAME and the purity of the biodiesel was 95.3%. In addition, Macías-Sánchez et al. [26] prepared biodiesel by in situ transesterification of microalgae biomass using sulfuric acid catalysis. After purification treatment with bentonite, the purity of biodiesel produced was increased only from 78.7% to 86.8%, which was much lower than the aquatic animal biodiesel synthesized in this study. The difference observed may have been due to the introduction of a co-solvent in this experiment, which increased the conversion efficiency of the oils and fats.

## 3. Experimental Section

### 3.1. Materials and Instruments

Fish tissue samples, i.e., grass carp, catfish, and sea bass waste fat and visceral tissue, were collected from the market in Fuzhou, China. Lard was prepared through laboratory extraction from commercially available pig fat (the extraction method is provided in Section 3.3), and was used to determine the fatty acid composition. FAME mixture standard with a mass concentration of 10 mg/mL was purchased from Sigma (Sigma-Aldrich; Shanghai, China). The organic solvents, i.e., n-hexane, chloroform, acetone, and methanol were analytical purity products, which were purchased from Thermo Fisher Scientific Co., Ltd. (Shanghai, China). Concentrated sulfuric acid (98.00 wt.%), NaCl, and other reagents were purchased from Sinopharm Chemical Reagent Co., Ltd. (Shanghai, China).

Agilent flame ionization detector-gas chromatography (FID-GC) 7890B with a 60 m AE.FFAP column (Lanzhou ZhongkeAntai Analytical Technology Co., Ltd.; Lanzhou, China) was used for lipid analysis. A FreeZone 4.5 L Desktop Freeze Dryer from Labconco Co., Ltd. (Kansas City, MO, America) was used for sample drying. The refrigerated centrifuge was from Thermo Fisher Scientific Co., Ltd. (Shanghai, China); constant temperature oven was purchased from Shanghai Shanzhi Instrument Equipment Co., Ltd. (Shanghai, China); constant temperature shaker was purchased from Anqing Jiejia Instrument Equipment Co., Ltd. (Anqing, China); and the automatic Kjeldahl nitrogen analyzer was purchased from Jinan Haineng Instrument Co., Ltd. (Jinan, China). 

### 3.2. Sample Preparation

First, the gallbladder, fish vesicles, and intestinal contents were separated from the fish waste samples. All these obtained contents were washed three times with tap water. These samples were blotted using filter paper and then freeze-dried for 6-7 h. After the dehydration process, these samples were homogenized in a homogenizer, while a small amount of tough tissue, such as the intestinal tissue, which could not be completely homogenized was discarded via a large-hole screen. The remaining homogenate biomass was evenly divided into several small containers and stored at −20 °C until use.

In order to improve the repeatability of this experiment, the total weight of each sample was sufficient for the whole experiment. Before experiment, we put a small amount of sample into a water bath with a constant temperature of 60 °C, heating for 8–15 min until the sample was completely melted; it was then shaken thoroughly.

### 3.3. Fat Extraction

Fat extraction experiments were conducted according to the method proposed by Nguyen et al. [27]. First, the biomass was mixed with n-hexane at a 1:10 (*m*/*v*) ratio; the mixture was then immersed in a constant temperature shaker (45 °C, 200 rpm) for 48 h. After being taken out, the mixture was vortexed and centrifuged at 2000 rpm for 5 min, and the supernatant was then extracted. Next, 1 mL of n-hexane was added into the precipitate, which was vortexed and centrifuged again at 2000 rpm for another 5 min, combining the supernatant. We then dried the supernatant with nitrogen, and the fat sample was kept in an oven at 70 °C until no further weight change was observed. Finally, the fat content, i.e., the mass percentage of the fat extracted, was calculated. The method used to determine the acid value of grass carp oil and its biodiesel was referenced to GB 5009.229-2016. The acid value of grass carp fish oil was extremely high, reaching 19.3 mg KOH g^−1^ oil, indicating that the free fatty acid content was very high, but the main component was still glyceride. This was the reason that transesterification was used in this work to express the synthesis of FAME.

### 3.4. In Situ Acid-Catalyzed Transesterification

The in situ transesterification was carried out in a completely sealed system, i.e., a 40 mL glass vial with PTFE septa. We first mixed 1 g of oily biomass with a certain amount of n-hexane; a pre-mixed H_2_SO_4_–methanol solution was then added and vortexed, and the system was then sealed using PTFE tape. The reaction occurred in an oven at a constant temperature, with the reaction vials being shaken every 20 min. When the reaction was completed, the temperature of vials was reduced to the room temperature. Next, 3 mL of saturated KCl solution was added, the whole solution was transferred to a 50 mL centrifuge tube, and 2 mL of n-hexane was used to rinse the original reaction vial, which was then combined into the centrifuge tube. After vortex mixing, the solution obtained was centrifuged at 2500 rpm for 5 min (the centrifuge rate and time were identical in Section 3.4); we then aspirated the supernatant, which contained biodiesel, into a 15 mL centrifuge tube. Subsequently 1 mL of n-hexane was added to the reaction solution again, and the procedure was repeated twice. The supernatant was collected into a 15 mL centrifuge tube, and 2.5 mL of distilled water was added to the extracted solution, followed by vortexing and centrifuging. Finally, the supernatant was extracted into another 15 mL centrifuge tube and about 0.5 g of anhydrous Na_2_SO_4_ was added to the solution, followed by vortexing and centrifuging, and the supernatant was aspirated; we added a further 1 mL of n-hexane to the centrifuge tube containing Na_2_SO_4_, which was vortexed and centrifuged again. In addition, the solution extracted was dried completely with a nitrogen gas flow. The biodiesel sample obtained after drying was crude biodiesel, and was further determined to a fixed volume of 10 mL with n-hexane. Next, 50 μL of it was accurately measured and added to a brown chromatographic vial, and the volume of solution was adjusted to 1100 μL with an extra n-hexane (equivalent to measuring the biodiesel to 220 mL). GC was then performed. The samples were finally stored at -20 °C. A schematic diagram of the procedure is given in Figure 6.

The purity of the biodiesel was determined by weighing 1.0 - 2.0 mg of biodiesel product, which was then dissolved with 1 mL n-hexane; the content of FAME was then analyzed by GC. Under the optimal reaction conditions, the residual solid waste of grass carp after the in situ transesterification was recovered by suction filtration, and the crude protein content of the solid sample was then determined using the Kjeldahl method (GB/T 6432-2018). The results were used to evaluate the feasibility of the byproduct for use in the preparation of animal feed. The conversion efficiency and purity of biodiesel were calculated using the following formulas.
(1)$$Conversion\;efficiency\;\left(\%\right)=\frac{m_{c}}{m_{r}}$$
(2)$$Purity\;\left(\%\right)=\frac{m_{c}}{c_{b}}$$
where *m_c_* represents the mass of FAME in crude biodiesel, *m_r_* represents the mass of fat contained in the raw material, and *c_b_* represents the mass of crude biodiesel.

### 3.5. Fatty Acid Derivatization Method

In this work, we made a modification to our previously proposed methods [28,29]. First, 10.0 mg of extracted fat was accurately weighed, and was then mixed with 1 mL of n-hexane. We pipetted 100 μL into an 8 mL of glass vial with PTFE septa and dried the solvent with nitrogen flow (to obtain 1.0 mg fat). Next, 2 mL of 2% H_2_SO_4_–methanol (*m/m*) solution was added, and the vial was sealed with PTFE tape and vortexed. The vials were placed in an 80 °C oven for 2.5 h, and were vortexed every 25 min. After completion of the reaction, the temperature was lowered, and an appropriate amount of saturated NaCl solution was added. Next, 2 mL of n-hexane was transferred into the solution, which was vortexed and centrifuged at 2500 rpm for 5 min. The supernatant was kept. Subsequently, 1 mL of n-hexane was added into the reaction mixture and the mixture was vortexed and centrifuged again at 2500 rpm for 5 min, combining the supernatant. The extract was dried with a nitrogen flow, and was then mixed with 1 mL of n-hexane for GC analysis. The fatty acid profile was calculated for different materials.

### 3.6. Lipid Assay

Agilent FID-GC 7890B with a 60 m FFAP column was used. Temperature programming: the oven temperature was increased from 100 °C to 250 °C at 7.5 °C/min, the final 250 °C was then held for 30 min. Helium was used as a carrier gas at a flow rate of 1.5 mL/min. The injector temperature was maintained at 260 °C and the injection volume was 2 μL.

### 3.7. Statistical Analysis

Each experiment was conducted in duplicate or triplicate, and the results were expressed as the mean ± standard deviation. In this work, the SPSS18.0 software was used for statistical analysis. Many studies have been conducted using the internal standard method [30] and external standard method [31] for FAME quantification. The results of the internal standard method are not affected by changes in the injection volume, so errors caused by changes in operating conditions are eliminated to a certain extent; however, the external standard method is more convenient and economical than the inner standard method. In addition, the accuracy of external standard method is comparable to that of the inner standard method if the injection reproducibility is high and the experimental operating conditions are stable. The GC used in this experiment was equipped with an accurate autosampler. We therefore applied the external standard method for the quantitative analysis of FAME. The FAME mixture standard with the concentration of 10 mg/mL was diluted to 7.5, 5.0, 2.5, and 1.0 mg/mL. The five concentration gradients were used as the external standard curve. We found that the R-squares of all standard curves were higher than 0.99, confirming the accuracy of the measured data.

## 4. Conclusions

In this study, technical conditions for production of biodiesel from a high acid value waste grass carp tissue through in situ transesterification were optimized with the use of a co-solvent. The optimal reaction conditions determined were as follows: 1 g grass carp tissue, hexane: methanol = 1: 9 (*v/v*), solvent dosage = 31 mL, H_2_SO_4_ loading = 0.6 mL, 75 °C, and 2.5 h reaction. We also prepared a byproduct with a crude protein content up to 34.73%, which has the potential to be used as a high-protein feed. The in situ biodiesel production technique could also be effectively applied to other fish processing wastes from freshwater fish and marine fish, preparing biodiesel with purity above 90%.

Our results showed that the in situ transesterification of aquatic animal waste tissue can be applied for production of biodiesel and high-protein feed. In future works, further research should be carried out on the purification measures, flammability, and economic evaluation of fish fat biodiesel. Expanded studies on in situ biodiesel production technologies (such as alkali catalysis, enzymatic catalysis, and supercritical methods) for aquatic animal tissue might further increase the biodiesel conversion efficiency, which is also an attractive research direction. In addition, the potential for waste residue from the in situ transesterification to be recovered for animal feed, with its high protein content, is also an effective way to reduce the price of biodiesel and should receive more extensive attention.

## Figures and Tables

**Figure 1 molecules-25-01650-f001:**
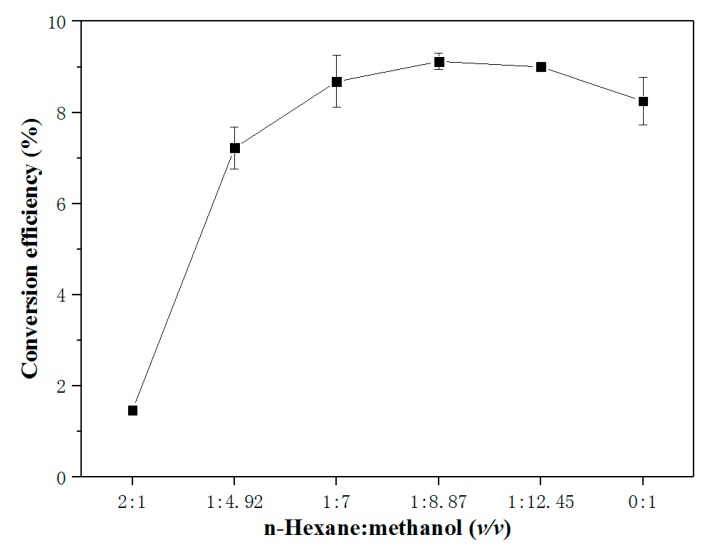
Effect of co-solvent to methanol ratio (*v/v*) on the in situ transesterification efficiency of grass carp tissue. The operational conditions were: 1 g grass carp tissue, solvent dosage (hexane + methanol) = 7.4 mL, 0.6 mL H_2_SO_4_, 1 h, and 85 °C.

**Figure 2 molecules-25-01650-f002:**
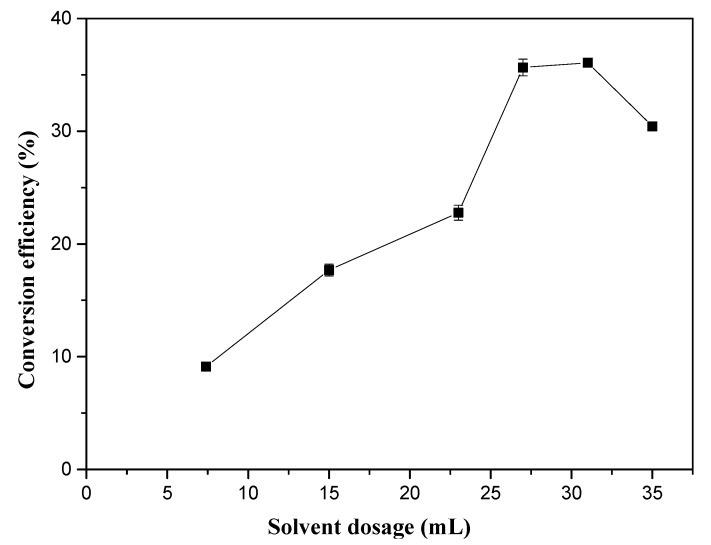
Effect of solvent dosage on the in situ transesterification efficiency of grass carp tissue. The operational conditions were: 1 g grass carp tissue, hexane: methanol = 1: 9 (*v*/*v*), 0.6 mL H_2_SO_4_, 1 h, and 85 °C.

**Figure 3 molecules-25-01650-f003:**
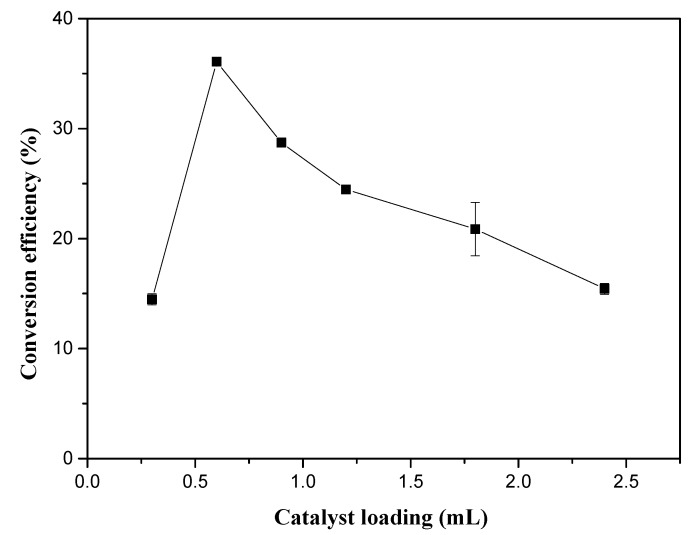
Effect of catalyst loading on the in situ transesterification efficiency of grass carp tissue. The operational conditions were: 1 g grass carp tissue, hexane: methanol = 1: 9 (*v*/*v*), solvent dosage = 31 mL, 1 h, and 85 °C.

**Figure 4 molecules-25-01650-f004:**
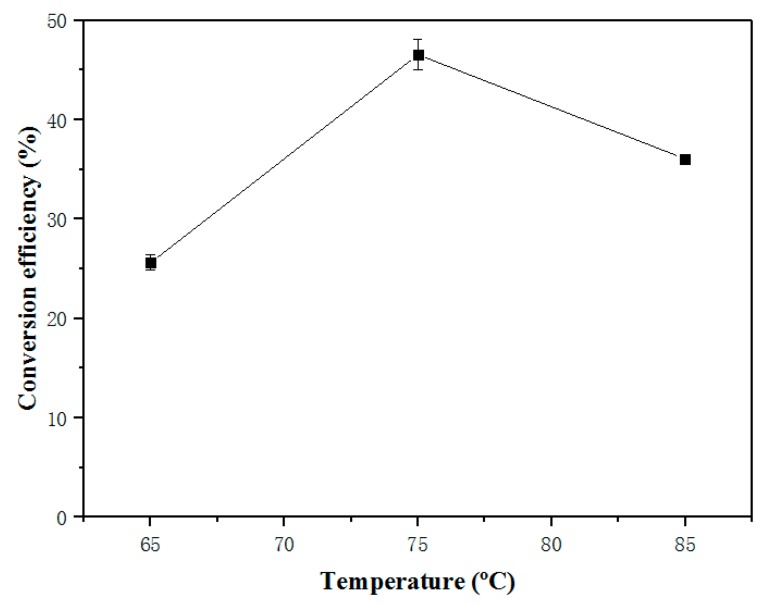
Effect of reaction temperature on the in situ transesterification efficiency of grass carp tissue. The operational conditions were: 1 g grass carp tissue, hexane: methanol = 1: 9 (*v*/*v*), solvent dosage = 31 mL, H_2_SO_4_ loading = 0.6 mL, 1 h.

**Figure 5 molecules-25-01650-f005:**
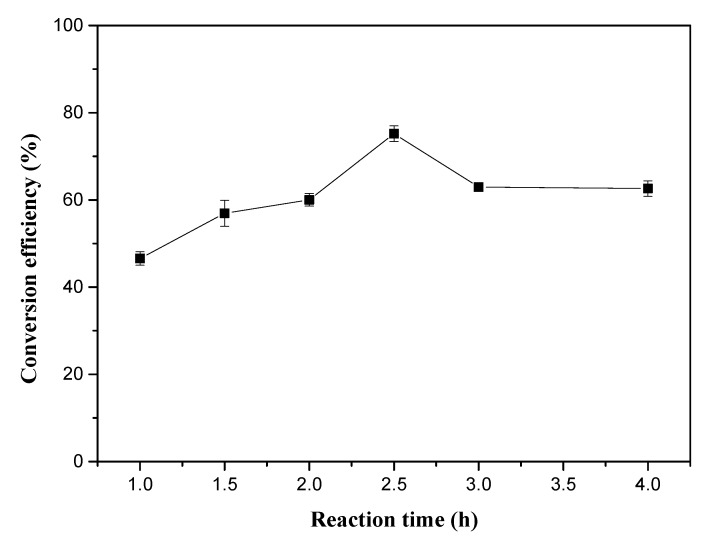
Effect of reaction time on the in situ transesterification efficiency of grass carp tissue. The operational conditions were: 1 g grass carp tissue, hexane: methanol = 1: 9 (*v*/*v*), solvent dosage = 31 mL, H_2_SO_4_ loading = 0.6 mL, 75 °C.

**Figure 6 molecules-25-01650-f006:**
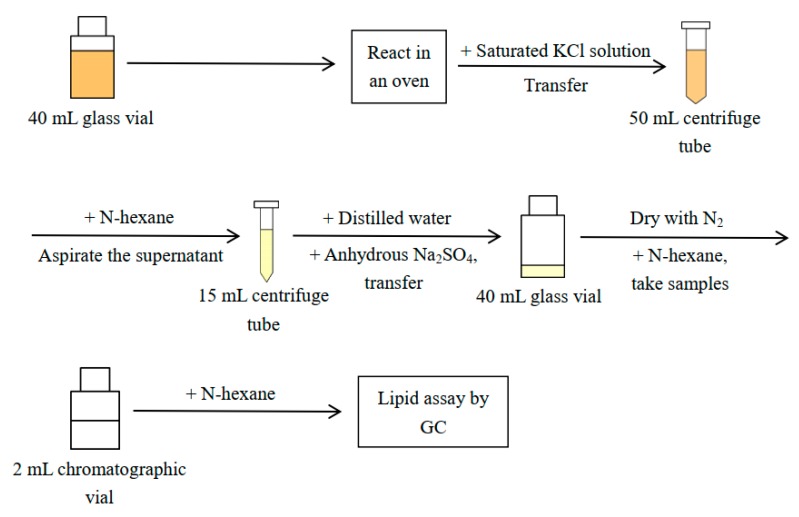
Schematic diagram of the procedure of in situ transesterification.

**Table 1 molecules-25-01650-t001:** Fatty acid profiles (%) of different fats used in this study.

Fatty Acid	Grass Carp Fat	Catfish Fat	Sea Bass Fat	Pig Fat
C6: 0	6.23	5.60	6.31	6.76
C14: 0	1.57	0.42	0.80	1.20
C16: 0	19.26	22.24	15.86	23.64
C16: 1	8.72	3.81	3.92	-
C18: 0	4.40	6.45	4.01	14.44
C18: 1	37.91	34.29	25.95	26.87
C18: 2	15.90	20.70	37.86	23.70
C18: 3	1.96	3.48	3.22	2.04
C20: 0	0.78	-	-	-
C20: 2	1.25	0.13	0.31	0.39
C21: 0	0.32	0.05	-	-
C20: 3	0.19	0.35	-	-
C20: 5	0.48	1.22	0.82	-
C22: 1	-	0.12	-	0.97
C22: 6	1.02	1.15	0.92	-
SFA	32.56	34.76	26.98	46.03
MUFA	46.63	38.21	29.88	27.84
PUFA	20.80	27.03	43.14	26.13

**Table 2 molecules-25-01650-t002:** In situ transesterification of different fish species’ tissue.

	Grass Carp Tissue		Catfish Tissue		Sea Bass Tissue
Conversion efficiency (%)		75.19		82.69	86.18
Purity (%)		91.99		93.84	100.00

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
