# Peer review of "Production of Biodiesel and High-Protein Feed from Fish Processing Wastes Using In Situ Transesterification"

_molecules, 2020, doi:10.3390/molecules25071650_

Round 1

Reviewer 1 Report

The subject of the paper clearly falls within the scope of this Journal.

The paper is very interesting, well written and well organized, and represents some advancement over the actual state-of-the-art. The ways and means are well described as well as the obtained results which are thoroughly discussed and conclusions are well drawn. The paper is also supported by some literature review. However, in order to make for a stronger paper, I suggest that the authors should cite and discuss the following relevant paper, which could be used as benchmark to the proposed approach:

Dias et al., “Review on biodiesel production processes and sustainable raw materials”, Energies, 12, 4408 (2019) - DOI: 10.3390/en12234408

I do recommend the publication of this paper, subjected to these changes.

Reviewer 2 Report

The work is well-written and has scientific merit.

There is some points that must be improved before publication:

Line 32: in situ should be in italic here and through the text

Line 106-107: “...research on the preparation of biodiesel from 106 such fish oil is still rarely reported...”. There is some reference here?

Line 121 and through the experimental section: please verify where the present form “are” should be replaced by “were”.

Line 121-122: Lard? Pig fat? The reason for this part is not clear to me.

Line 133: What dehydration process was used?

Line 144: It would be more appropiate Nguyen et al. (2017)

Line 152: What “GB 152 5009.229-2016 means”?

Line 161: How much is “certain amount”? What is the proportion of each component in the sulfuric: alcohol mixture

Line 194. After the description, the insertion of an schematic figure would be very interesting.

Line 279: Just “(2018)”

Line 369 and 413: Just “(2009)”

Reviewer 3 Report

Zhang et al show results concerning the preparation of biodiesel using in-situ transesterification with aquatic animal tissue, using the grass carp (Ctenopharyngodon idellus) biomass as a representative to optimize the reaction conditions.

Interestingly, the authors reached a biodiesel purity reachesup to 100% for the sea bass wastes, which is higher than 96.5% in coparison with the EN 14214-2008. The in-situ method proposed here potentially saves large cost of biodiesel production compared to the conventional ways, which usually require the high-cost pretreatment of raw materials.

I agree that this study willinspire new strategies to prepare biodiesel and high protein feed simultaneously from the aquatic animal biomass using the novel in-situ transesterification.

For theses reasons, I accept this paper in the present form.

Yours sincerly

Author Response

We are very pleased that you highly value our research . Thank you!

Reviewer 4 Report

The manuscript by Zhang et al. reports on the development of a technique to utilize in situ transesterification to prepare biodiesel and high protein feed from aquatic animal biomass using grass carp (Ctenopharyngodon idellus) as the model.This is a careful study on identifying the optimal parameters for in situ transesterification. The authors were then able to successfully apply this technique to other aquatic animal tissues.

Specific Comments/Questions:

(1) There are few grammatical and typographical errors scattered throughout the manuscript.

(2) The Experimental Section needs to be written in more detail. It would be difficult for anyone outside the lab to follow this in situ transesterification protocol. For example, the brand of the instruments (e.g., centrifuge, GC, etc.) need to be provided. The speed of centrifugation (in rpm) is given for some steps, but not others. The centrifuge speed (in x g) needs to be provided for each of the steps.

(3) Were the sample preparation and fat extraction steps also optimized for the grass carp tissues?

(4) In the in situ transesterification protocol, were higher temperatures or longer times attempted? Previous studies had used between 90-120 degrees and 12 hours (versus maximum of 4 hours used in this study). Unlike the graphs of the other parameters that showed a marked decrease from the optimal, I could argue that both the temperature and tine parameter curves are plateauing. Did the authors attempt dramatically higher temperatures or extended times?

(5) Figure 6 can be removed. It does not fit where cited in the Results and Discussion and does not add to the manuscript.
